# How Lineage Tracing Studies Can Unveil Tumor Heterogeneity in Breast Cancer

**DOI:** 10.3390/biomedicines10010003

**Published:** 2021-12-21

**Authors:** Elena Vinuesa-Pitarch, Daniel Ortega-Álvarez, Verónica Rodilla

**Affiliations:** Cancer Heterogeneity and Hierarchies Group, Josep Carreras Leukaemia Research Institute (IJC), 08916 Badalona, Spain; evinuesa@carrerasresearch.org (E.V.-P.); dortega@carrerasresearch.org (D.O.-Á.)

**Keywords:** breast cancer, lineage tracing, heterogeneity, cellular plasticity

## Abstract

Lineage tracing studies have become a well-suited approach to reveal cellular hierarchies and tumor heterogeneity. Cellular heterogeneity, particularly in breast cancer, is still one of the main concerns regarding tumor progression and resistance to anti-cancer therapies. Here, we review the current knowledge about lineage tracing analyses that have contributed to an improved comprehension of the complexity of mammary tumors, highlighting how targeting different mammary epithelial cells and tracing their progeny can be useful to explore the intra- and inter-heterogeneity observed in breast cancer. In addition, we examine the strategies used to identify the cell of origin in different breast cancer subtypes and summarize how cellular plasticity plays an important role during tumorigenesis. Finally, we evaluate the clinical implications of lineage tracing studies and the challenges remaining to address tumor heterogeneity in breast cancer.

## 1. Introduction

Breast cancer is the most common cancer in women worldwide and is the second leading cause of cancer death in women, exceeded only by lung cancer (based on data from the World Health Organization, 2021). This type of cancer originates in the mammary gland, which is a ductal tree composed of two epithelial compartments: cells facing the ductal lumen called luminal cells (LCs), and basal cells (BCs) found in the outer layer with a capacity to contract, which includes basal progenitor cells and terminal differentiated myoepithelial cells. Luminal cells can be further subdivided into two independent subpopulations based on the expression of the hormone receptor, estrogen receptor alpha (ERα) [1] (Figure 1).

In clinical practice, breast tumors are classified based on the histological expression of ERα, progesterone receptor (PR), receptor tyrosine-protein kinase ErbB-2 (HER2), and the proliferation marker Ki-67. They are divided into three main groups: hormone receptor-positive subtype, which includes tumors expressing ERα and/or PR, which are subclassified as luminal A or B depending on the percentage of Ki-67; HER2-positive tumors, defined by the presence of *ERBB2*/HER2 amplifications and loss of ERα expression; and triple-negative breast cancer (TNBC), characterized by the lack of expression of the aforementioned molecular markers [2] (Figure 2).

At the cellular level, breast cancer is a heterogeneous disease [3], thereby evoking disparate phenotypes not only between patients, but also within the tumor itself, known as inter- and intratumor heterogeneity, respectively [4]. It was suggested that this intratumor heterogeneity is perpetuated by tumor stem-like cells, leading many research groups to focus on the identification of biomarkers that allow them to target those cells responsible for tumor maintenance, including CD44 [5], CD29 and CD49f, [6], CD133 [7], Lgr5 [8], Procr [9], ALDH [10], or CD61 [11]. Although the origin of breast cancer, as many other types of cancer, remains largely elusive, a plausible theory is that adult mammary stem/progenitor cells, which are very long-lived compared with differentiated cells, are better targets for accumulating the multiple genetic mutations necessary for malignant transformation [12]; however, accumulating evidence has demonstrated that specific mutations in differentiated cells are also able to initiate a tumor [13,14,15,16]. Importantly, both luminal and basal cells are possible targets for malignant transformation [17,18].

During the last decade, lineage tracing studies have provided deeper insights into the cellular heterogeneity and molecular mechanisms underlying cellular plasticity in different subtypes of breast cancer. Accordingly, this review outlines the main milestones concerning this technology in the breast cancer context, as well as underlining unsolved questions and future prospects in the field.

## 2. Lineage Tracing Is the Gold-Standard Approach for Exploring Cellular Hierarchies and Tumor Heterogeneity in Breast Cancer

Genetic fate mapping studies commonly use a lineage-specific promoter followed by an inducible form of Cre recombinase fused to the mutant ligand-binding domain of the human estrogen receptor (CreER, CreERT, or CreERT2) [19,20,21], which does not bind to endogenous estradiol, but can be activated by the administration of synthetic ligands, such as tamoxifen or 4-hydroxytamoxifen (4-OHT). Furthermore, this Cre line needs to be mated with a reporter line carrying the β-galactosidase enzyme [22], one or several fluorescent protein(s) [23,24,25,26,27], or barcoding sequences [28,29], in order to monitor the targeted cells. In this way, tamoxifen or 4-OHT drive the inducible Cre to the nucleus, where it promotes the recombination of *loxP* sequences, allowing the expression of the reporter (Figure 3A). The use of this particular system has been controversial, since high doses of tamoxifen can delay mammary gland development [30]; however, the use of low doses (0.1 mg/g of mouse body weight) is a good balance between correct mammary epithelial development and sufficient labeling efficiency [31]. Alternatively, other studies have opted for tetracycline-inducible systems, where tetracycline administration, or its analogue doxycycline (Dox), enables the expression of a reverse tetracycline-controlled transactivator (rtTA). Activated rtTA will bind to tetracycline response elements (TRE), which trigger the expression of Cre recombinase to mediate the recombination of *loxP* sequences located at the reporter transgene [32,33] (Figure 3B). Advantageously, this system allows us to perform lineage tracing at saturation [34], relying on the long-term administration of Dox and inducing reporter recombination in every single cell of a given lineage, overcoming the low recombination efficiency obtained by classic lineage tracing experiments using ER-dependent Cre lines.

A new genetically engineered mouse model using the Flippase (Flp)/flippase recognition target (*Frt*) system was developed to specifically study the mammary gland [35]. The authors generated a transgenic mouse line expressing a mouse codon-optimized Flp under the control of the mouse mammary tumor virus (*MMTV*) promoter, which recombines *Frt* sequences exclusively in mammary epithelial cells. Similar to the doxycycline-dependent system, this in vivo strategy does not require the use of tamoxifen for activating Flp enzyme. Importantly, this model consists of a Cre alternative recombinase (Flp), which allows the combination of multiple site-specific recombination systems, such as Cre/*loxP* and Flp/*Frt* [35,36].

One of the most suitable approaches for lineage tracing is pulse experiments, where 16–24 h after Cre recombination, the targeted cells are analyzed; and chase experiments, which allow us to visualize the progeny of the targeted cells, genetically labeled regardless of the expression of the specific gene used for lineage tracing. Recently, some groups have used this technology to explore the capacity of different mammary epithelial cells to initiate tumors or metastasize, as well as to understand the potential mechanisms underlying cellular plasticity within breast tumors, which could ultimately lead to new therapeutic strategies for treating heterogeneous tumors in the near future.

### 2.1. Searching for the Tumor Cell of Origin in Different Breast Cancer Subtypes

Gene expression analyses have provided a new sub-classification of breast cancer patients based on their transcriptomic profile, which allowed them to be originally subclassified into five different molecular breast cancer subtypes: luminal A, luminal B, HER2-positive, normal-like, and basal-like [37]. Further gene expression studies identified a new breast cancer subtype known as claudin-low or mesenchymal-like [38]; and later, clustering analysis of genomic and transcriptomic data of breast tumors revealed 10 novel integrative clusters, which displayed different copy number alterations and gene expression profiles associated with distinct clinical outcomes [39].

Importantly, several studies compared the gene expression signature of different healthy mammary epithelial cells (MECs) with transcriptional profiles of different tumor subtypes [17,40,41]. Interestingly, it was found that the luminal progenitor cell signature resembled that of basal-like tumors, suggesting that the cell of origin in this breast cancer subtype could be an LC, both in mouse and human [17,40,41]. Conversely, the BC signature was upregulated in the claudin-low subtype, and the mature luminal gene signature was closely aligned with the luminal A and B subtypes [17,41]. Overall, these results introduced comparative expression profiling as a powerful tool to elucidate the cell of origin in different cancer subtypes, which could serve as a cellular target for oncogenic events; however, only genetic studies at the single-cell level or lineage tracing experiments are the current definitive approaches to identify the cell of origin in each breast cancer subtype.

So far, only a few studies have explored this particular issue using lineage tracing tools. A good example is a recent study that proved that cells positive for leucine-rich repeat-containing G-protein coupled receptor 6 (*Lgr6*), which is expressed in LCs and BCs during the early stages of tumorigenesis, contributed to mammary tumor progression [42]. Specifically, when Lgr6^pos^ cells were genetically labeled at the hyperplasia stage (P12, 12-day-old) in a mouse mammary tumor virus promoter-driven Polyomavirus middle T antigen breast cancer mouse model (*MMTV*-PyMT), which mostly generates luminal tumors, these cells clonally expanded, contributing to the formation of carcinomas [42]. Surprisingly, when they used the Medroxyprogesterone Acetate (MPA) plus 7,12-Dimethylbenz[a]anthracene (DMBA)-dependent model to generate mixed luminal and basal tumors [42], their lineage tracing analysis showed that neither basal nor luminal Lgr6^pos^ cells were involved in the formation of basal-like tumors. These results suggest that Lgr6^pos^ cells could be the cell of origin in the luminal breast cancer subtype exclusively; however, the fact that the *Lgr6* promoter is activated in LCs and BCs at the start point of the lineage tracing makes it complicated to draw strong conclusions regarding the true cell of origin in each breast cancer subtype. Remarkably, when *Lgr5*-expressing cells (exclusively expressed in BCs [43]) were traced in C3(1)Tag mice, a murine model that spontaneously develops TNBC [44,45], the hyperplastic lesions generated were mostly Lgr5^pos^-derived progeny, denoting Lgr5^pos^ cells as the cellular origin of TNBCs [46]; nevertheless, an exhaustive histological characterization of the resulting tumors was missing in this work, making it difficult to conclude which specific types of tumors were formed from these cells.

Fascinatingly, a recent work combined lineage tracing analysis and the RCAS-TVA system. This consists of expressing the TVA cell surface receptor under the control of a specific target promoter recognized by an avian leukosis virus-derived vector (RCAS) [47] to elucidate the contribution of ERα during cancer progression and metastasis of HER2-positive tumors [48]. By using *Esr1*-Cre/*MMTV*-TVA/*Rosa*26-tdRFP mice infected with RCAS-*Erbb2* (to generate HER2-positive tumors), the authors could demonstrate that ERα^pos^ tumor cells with an overexpression of HER2 have to progressively lose their ERα expression in order to clonally expand and metastasize [48]. Importantly, HER2-positive cells originated from ERα^pos^ cells were more aggressive than those originated from ERα^neg^ cells, suggesting that the cell of origin plays an important role in the clinical outcome of breast cancer patients.

Undoubtedly, more studies are required to figure out which mammary epithelial cell type is the cell of origin for each specific tumor type. The lack of good preclinical breast cancer mouse models that truly recapitulate human subtypes, and the selection of suitable inducible Cre lines, which exclusively label one mammary epithelial cell type at a time, are the major concerns that the research community currently face to address this important issue.

### 2.2. Importance of the Epithelial-to-Mesenchymal Transition during Tumor Progression

The epithelial-to-mesenchymal transition (EMT) is a biologic process by which epithelial cells lose their cell polarity and cell–cell adhesion to undergo multiple biochemical changes that enable them to assume mesenchymal attributes, such as an elongated shape, fibroblast-like morphology, enhanced mobility, invasiveness, resistance to apoptotic stimuli, and production of extracellular matrix components [49]. Crucially, this process could be exploited by tumor cells, allowing them to detach from each other within the primary tumor and metastasize to distant organs [50].

In vivo monitorization of different mesenchymal markers (FSP1 and Vimentin) using different breast cancer models demonstrated that the EMT process does not contribute to lung metastasis development [51]. These studies evaluated the appearance of EMT in two breast cancer models (*MMTV*-PyMT and *MMTV*-Neu), being a biological process that can be observed in vivo; however, the metastases developed were neither FSP1- nor Vimentin-derived progenies [51,52,53]. Notwithstanding, FSP1-derived mesenchymal cells can undergo mesenchymal-to-epithelial transition (MET) and contribute to tumor recurrence, although this was addressed using serial transplantations as a rough way of recapitulating this tumor process [54]. The main concern regarding these studies was that these murine lines were targeting a small fraction of the total cells that undergo EMT, having a non-negligible difference between the total percentage of mesenchymal E-Cadherin (Ecad)^low^ cells and FSP1^pos^-cells (5% and 0.3%, respectively) [52]. Crucially, when Ecad^low^ tumor cells were injected into the circulation, these cells generated metastases [52]; however, these experiments did not address the fact that this process occurs naturally in vivo during the progression of this disease.

The fact that many different cells can undergo EMT and the lack of a universal marker encouraged the combinatorial design of Cre and Dre systems to study this biological process. Recently, Li and colleagues generated *EMTracer*, a triple transgenic mouse model carrying *Kit*-CreER, *EMT*gene-LSL-Dre, and NR1-reporter. Thus, this *EMTracer* was crossed with the *MMTV*-PyMT model to monitor EMT during tumor progression [55]. With this strategy, after tamoxifen administration, luminal-*Kit*^pos^ cells recombined LoxP sequences expressing ZsGreen fluorescent protein, inducing Dre expression exclusively in mesenchymal cells which were positive for *Vim* or *Cdh2*/N-cadherin, turning *tdTomato* positive [55]. In their functional assays, they found that Vimentin was not functionally required to metastasize (as previously reported); however, the activation of N-cadherin was critical for lung colonization [55]. In addition, the authors demonstrated that breast cancer cells underwent the EMT program during primary tumor growth rather than during dissemination or lung colonization [55], indicating the importance of studying EMT at different stages of the metastatic cascade to draw reliable conclusions.

The heterogeneity underlying the EMT process could be explained by the emergence of a partial rather than a full EMT [56], resulting in the appearance of intermediate hybrid states that share epithelial and mesenchymal features. In fact, combining lineage tracing strategies with single cell RNA-sequencing (scRNA-seq) analyses at different time points of the metastatic cascade could shed light on the characterization of the different EMT transitioning states that breast cancer cells undergo.

### 2.3. Differential Clonal Expansion during Tumor Progression and the Metastatic Process

New microscopy technologies, such as intravital or 3D-whole mount imaging, have allowed the visualization of mammary tumor progression from adenoma to carcinoma in vivo and at large-scale single-cell resolution, respectively [57,58]. Combining intravital imaging with random multi-labeling of the mammary gland (using *Rosa26*-CreERT2), Zomer and colleagues observed that only a small subset of tumor cells clonally expanded during tumor progression, and the vast majority of cells within the primary tumor either disappear, grow slowly, or partially expand to finally regress [57]. The combination of these technologies with cell-specific lineage tracing analysis to further study the potential of each MEC to metastasize still represents a gap in our field of knowledge.

Performing multicolor fluorescent lineage tracing in the *MMTV*-PyMT model demonstrated that the metastatic process is produced by the collective dissemination of cancer cells forming cohesive clusters rather than the serial seeding of single tumor cells [59]. Concretely, after the orthotopic implantation of mammary tumors formed of cells randomly labeled with different fluorescent proteins, the resulting lung metastases were formed of multicolored cells, signifying multicellular seeds [59]. Moreover, these lineage tracing studies showed multicolored tumor cell clusters at five different stages of metastasis: collective invasion, locally disseminated clusters in the adjacent stroma, extravasated tumor emboli, circulating tumor cell clusters, and distant micro- and macro-metastases [59].

A powerful approach is combining lineage tracing with scRNA-seq by introducing genetic barcodes. Indeed, using this strategy, Ginzel and colleagues measured the tumorigenic capacity of different oncogenic HER2 isoforms (HER2, d16HER2 and p95HER2) within the same mammary gland [60]. Specifically, the authors mated *MMTV*-Cre mice with HER2-Crainbow (HER2BOW) mice, which encoded for the three HER2 variants, fluorescently barcoded and flanked by *LoxP* sites. Using fluorescence and RNA sequencing, they could characterize the tumor phenotype associated with each barcoded isoform [60]. Although wild-type HER2 rarely induced indolent tumors, d16HER2 generated luminal-like proliferative in situ lesions, which eventually progressed, and p95HER2 prompted the early appearance of invasive cancers characterized by double-positive luminal and basal epithelial cells. From a clinical standpoint, these results underscore the importance of subclassifying HER2-positive breast cancer patients based on their HER2 isoform [60].

## 3. In Vivo Models to Study Mammary Gland Tumorigenesis

Many groups have tried to recapitulate the wide diversity of breast cancer subtypes detected in the clinics, designing preclinical mouse models that overexpress oncogenes or silence tumor suppressor genes. Indeed, recent studies have relied on Cre/lox systems for defining novel drivers of mammary tumorigenesis and assessing their consequences in different cellular contexts.

### 3.1. Classic Preclinical Models to Recapitulate Different Breast Cancer Subtypes

It was reported that specific murine strains can spontaneously develop breast tumors, such as CH3, that produces adenocarcinomas with a latency of 6–10 months [61]; BALB/c that also generates adenocarcinomas at 12 months of age [61]; and the Kunming strain that can develop invasive ductal carcinomas in 13.5 months [62]. In addition, there are inducible models that can be sorted by chemical treatments (DMBA or N-nitroso-N-methylurea (NMU)) [63]; physically, by the effects of radiation [64]; or biologically, by lentiviral infection [65]. However, the transcriptomic profiles of most of these models have not been fully analyzed. In order to shed light on this important issue, different research groups analyzed in depth tumors derived from different animal models by comparing their gene expression profiles with different human public databases, and calculating to what extent these models resembled the human disease [45,66]. Thus, murine models were divided into mesenchymal (also known as claudin-low subtype), basal, luminal, or HER2-enriched tumors. Among the animal models with tumors presenting mesenchymal features, mainly characterized by the expression of *Vim* and *Snai1*, they found the *MMTV*-Cre/Brca1^Co/Co^/p53^+/−^ [67]; DMBA-induced [68], few C3(1)-Tag [44], *MMTV*-Lpa [69], WAP-T_121_ [70], and p53^+/−^ irradiated [71] models. Recapitulating human basal-like tumors, they included *MMTV*-Cre/Brca1^Co/Co^/p53^+/−^ [67], DMBA-induced [68], C3(1)-Tag [44], *MMTV*-Myc [72], *WAP*-Myc [73], *WAP*-Tag [74], *MMTV*-Aib1 [69], and *MMTV*-Wnt1 [75] mice. Resembling luminal tumors, they detected *MMTV*-Neu [76], *MMTV*-PyMT [77], *WAP*-Myc [73], *MMTV*-Myc [72], *MMTV*-Aib1 [69], *MMTV*-Hras [72], and WAP-Int3 [78]. Additionally, recapitulating HER2-enriched tumors, they found *MMTV*-Neu [76], Bgr1^+/−^ [79], p18^−/−^ [80], Rb^−/−^ [81], *MMTV*-Aib1 [69], *WAP*-Cre/Etv6 [82], *WAP*-T_121_ [70], and *MMTV*-Fgf3 [83]. (Table 1).

Some of these models have been the in vivo approaches of reference to study breast cancer for decades; however, they are not sufficient to understand the wide spectrum of human breast tumors. Moreover, the major differences in the composition of the stroma between the murine and human mammary gland, being more adipocytic and less fibrotic in mice than in humans [84], might play an important role during tumor progression, as well as the colonization of different metastatic organs, since mouse models only develop distant lung metastases, whereas humans are able to metastasize into lung, liver, bone, or brain [85].

### 3.2. Cellular Plasticity Plays an Important Role in the Development of Mammary Tumors

Numerous lineage tracing studies in healthy mammary glands have demonstrated that there are no multipotent stem cells in adult mice, but distinct pools of unipotent stem cells, which self-sustain the lineage restriction of each mammary epithelial population [86,87,88]. However, adult MECs have been shown to be extremely plastic under different stress situations, such as transplantation [86], oncogene activation [18,43,89,90], cellular ablation [91], or the ectopic expression of key cell fate determinants [92,93], interconverting their cellular potency and destiny. This cellular plasticity observed in normal epithelial cells may be conceivably magnified in tumors, thus contributing to the cellular heterogeneity observed in breast cancer.

In the cancer context, two independent groups have shown that luminal ERα^pos^ tumors can arise from BCs and LCs due to the expression of the oncogenic form PIK3CA^H1047R^ [43,89]. In fact, PIK3CA^H1047R^ was enough to induce cell plasticity and the acquisition of multi-lineage features, defined by the expression of both luminal and basal gene signatures at the same time [43]. Interestingly, when the same oncogenic hit was overexpressed in LCs (*Krt8*-expressing cells) or BCs (*Lgr5*-expressing cells), the resulting tumors were basal-like, HER2-positive, and luminal B (*Krt8*-derived tumor), or mainly luminal A and B (*Lgr5*-derived tumors) [43] (Figure 4). Similar results were found using the *Krt5* promoter to target BCs [89]; concretely, overexpression of PIK3CA^H1047R^ in BCs led to the formation of luminal B tumors (Figure 4), while *Krt8*-expressing cells generated luminal B and basal-like tumors [89]. All these results suggest that ERα-positive tumors with mutations in PI3K could originate from LCs and also BCs, whereas luminal ERα-negative, basal-like, and HER2-enriched tumors may exclusively arise from LCs. Indeed, these studies show how different the cellular plasticity is in LCs and BCs, and how the activation of a specific oncogene can give rise to different latency and tumor types depending on the cell of origin.

In the same direction, the loss of *Brca1* and *Tp53* in different cellular compartments resulted in the development of different breast cancer subtypes [18]. In this particular work, the authors used *Krt14*-Cre or *Blg*-Cre lines to monitor BCs or ERα^neg^ LCs, respectively, and mated them with Brca1^fl/fl^/p53^+/−^ mice. Importantly, BRCA1/p53 deficiency generated different types of tumors that histologically expressed basal markers (Keratin-14 or p63) and luminal markers (Keratin-18 and/or ERα), with similar but not identical gene expression profiles, which closely resembled that of human basal-like tumors, regardless of the cell of origin (Figure 4). Importantly, only *Bgl*-derived tumors histologically resembled human BRCA1 loss-of-function [18].

Strikingly, using the luminal Cre line *Wap*-Cre, different groups were able to observe a luminal-to-basal conversion, either overexpressing NTRK3 [94], active NOTCH1 (N1ICD) [95], or KRAS^G12D^ [96]. NTRK3 was able to induce mixed tumors bearing both basal and luminal cells, as well as hybrid tumors with cells expressing basal and luminal markers simultaneously [94]; N1ICD specifically generated tumors that transcriptionally matched with distinct luminal subtypes, but also with a mixed subtype containing luminal and basal identities [95], and the exogenous expression of mutant KRAS^G12D^ led to metastatic claudin-low mammary tumors with a mesenchymal-like phenotype [96] (Figure 4).

**Figure 4 biomedicines-10-00003-f004:**
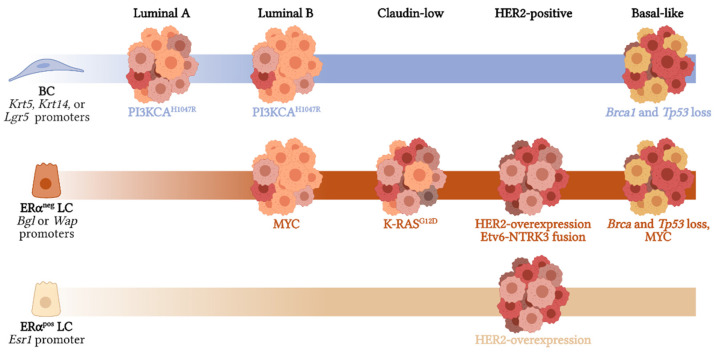
Schematic representation of the mammary cellular plasticity during tumorigenesis. The use of different murine Cre lines (*Lgr5*-Cre^ERT2^, *Krt5*-Cre^ERT2^ or *Krt14*-Cre to target BCs; *Bgl*-Cre or *Wap*-Cre to trace ERα^neg^ LCs; and *Esr1*-Cre to monitor ERα^pos^ LCs) to induce the expression of specific mutations in different mammary epithelial cell types resulted in the generation of tumors that resembled some of the human subtypes: luminal A [43], luminal B [43,89], HER2-enriched [48,66], claudin-low (or mesenchymal-like) [96], and basal-like [18,73] tumors. Notes: this figure only includes studies that have used comparisons with human breast cancer transcriptomic datasets (PAM50, Hu306 or similar); studies performed with non-specific Cre lines, such as *MMTV*-Cre or *Krt8*-Cre^ERT2^ (which labels all MECs, or both LC populations, respectively), were discarded. This figure was created with Biorender.com (accessed on 12 December 2021).

Another interesting strategy to study cellular plasticity is the use of the RCAS-TVA system [47]. Using this retrovirus-mediated in vivo lineage tracing, Hein and colleagues engineered RCAS vectors bearing either PyMT or HER2 constructs, which were produced and infected in *Wap*-TVA transgenic mice that specifically express the TVA receptor in LCs [90]. Similar to the aforementioned studies, the activation of certain oncogenes in the luminal compartment was sufficient to develop tumors with either luminal, basal, or mixed (Keratin-8 and Keratin-5-positive) features [90].

Collectively, these studies have demonstrated that cellular plasticity could be responsible for the intra- and inter-tumor heterogeneity found in breast cancer and pose new questions such as whether any mammary epithelial cell has the potential to become a tumor-initiating cell, or, by contrast, only a suitable combination of oncogenic hits is the crucial determinant in developing a tumor regardless of the cell of origin. Hence, these findings emphasize the importance of searching for key factors underlying cellular plasticity, which could have significant implications for cancer therapeutics.

## 4. Clinical Implications of Lineage Tracing Studies and Future Perspectives

Clinical decisions are made depending on the molecular subtype diagnosed, which includes anti-hormone therapies for those patients with ERα^pos^ cells, anti-HER2 treatments for HER2-positive patients, and chemotherapy for patients diagnosed with TNBC, due to the lack of specific targets [2]. These therapeutic strategies do not take into account that tumors are heterogeneous, meaning that they are composed of different types of cells, which could be the main cause of clinical failures. In this sense, the main handicap that researchers are currently facing is the lack of reliable preclinical models that resemble this human intratumor heterogeneity. Currently, the vast majority of in vivo mammary tumorigenesis models available depend on the expression of a specific driver, which eventually contributes to the formation of practically (intra)homogeneous tumors. The combination of multiple in vivo systems (Cre/*lox*, RCAS/TVA, Dre/*rox*, and Flp/*Frt*) could be the key to designing a model able to recapitulate the tumor evolution observed in human breast cancer.

Remarkably, lineage tracing studies have disclosed that the cell of origin matters. Here, we have presented several good examples, such as the work of Ding and colleagues that proved that HER2-positive tumors with an ERα^pos^ or ERα^neg^ cellular origin will determine their aggressiveness and metastatic capacity [48]. Moreover, these in vivo approaches have also revealed an inherent cellular plasticity in MECs; along this line, different groups have demonstrated that upon the activation of a specific oncogenic hit, MECs can acquire new transcriptomic features and expand to develop tumors with different cell fate signatures [18,43,89,94,95,96]. Some of these studies tried to understand whether the cell of origin or the oncogenic activation were playing a major role during tumorigenesis, and concluded that the cell of origin was crucial for determining the tumor subtype and/or aggressiveness; however, in all cases, those specific oncogenes (PI3KCA [43,89] or HER2 [48]) were able to generate tumors, suggesting that both facts (cell of origin and oncogenic hit) are equally important. Numerous studies on the normal mammary gland have demonstrated the high plasticity of BCs upon different stressors (transplantation or cell ablation [86,91]) compared with LCs; nevertheless, in the cancer context, all these studies demonstrated that LCs are more plastic than BCs, being able to generate a wider range of breast tumor subtypes [18,43,89,94,95,96], representing a new field to be explored.

Human cells contain somatic mutations that have served as genetic barcodes to perform retrospective lineage tracing analysis in healthy and diseased human tissues [97]. For example, topographic single-cell sequencing from laser-capture microdissected breast cancer samples at different tumorigenic stages, analysis of copy number alterations and clonal dynamics of different areas suggested a multiclonal invasion model for breast cancer [98]. Another strategy is the use of mitochondrial DNA mutations as a natural barcode, which McDonald and colleagues used to find common mutations that would indicate a common cell of origin in both normal and premalignant breast sections [99]. Alternatively, the use of DNA barcodes, introduced by infection in isolated normal human mammary cells, revealed a complex clonal landscape within heterogeneous breast tumors expressing KRAS^G12D^ [100].

Notably, over the last years, single-cell RNA sequencing has emerged as a new tool to replace lineage tracing studies, since it provides a recapitulation of the clonal dynamics of different cell populations within a tumor in a retrospective manner; however, the question of the identity of the tumor-initiating cell still remains to be solved with this type of technology. Combining both approaches would allow us to genetically and phenotypically trace each individual cell, redefining the phylogenetic trees, cell trajectories, and cell–cell interactions. In keeping with recent breakthroughs, the development of barcode systems has enabled us to target individual cells with unique nucleic acid sequences [101]. This technology could be used for lineage tracing together with the sc-RNAseq technique to reveal the transcriptomics of each cell population within heterogeneous mammary tumors, and also to perform high-throughput genetic screening to discover key plasticity factors and tumor drivers, being potential druggable targets for breast cancer therapeutics [102]. For example, Ying and Beronja employed long-term lineage tracing using stable barcodes to study mammary tissue hierarchy, ensuring each progenitor was labeled with a single barcode [102], and designed a large-scale genetic screening with a barcoded lentivirus library that targeted multiple clinically relevant mutations [102]. Definitively, the combination of multiple technologies will allow us to identify new crucial biomarkers and novel therapeutic targets, especially for TNBC patients who have no targeted therapies available.

Beyond lineage tracing, multi-omic and high-throughput methods will help us to understand the complex genomics that underlie the human cancer disease. Indeed, single-cell genomics has enabled us to associate specific genetic mutations with different molecular subtypes [103], and single-cell transcriptomics has provided cancer-specific gene signatures that could help clinicians to determine the prognosis of patients [104]. For instance, there are currently different gene expression profiling tests for breast cancer in clinics, such as MammaPrint^®^ and Oncotype DX^®^, both of which predict the risk of distant disease recurrence. Moreover, among some spatial transcriptomic technologies, fluorescence in situ hybridization (FISH) methods are commonly used in cancer diagnosis as they allow the detection and chromosomal location of specific genes which are aberrantly expressed or harbor rare mutations in tumors [105]. On the other hand, there are other techniques such as laser capture microdissection and photoactivatable transcriptome in vivo analysis that could be combined with sequencing-based approaches to decipher the genetic information of the desired area of the tissue [105]. The integration of all these spatial data promises the identification of multiple reliable molecular biomarkers that would be helpful for the diagnostics and therapeutics of breast cancer patients in the near future.

## Figures and Tables

**Figure 1 biomedicines-10-00003-f001:**
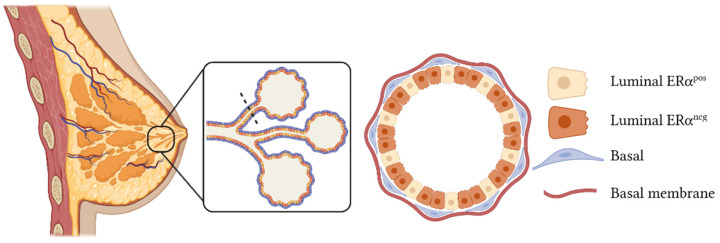
Model of normal mammary gland structure. This tissue is composed of ducts, which are formed by three epithelial populations: basal cells, in contact with the basal membrane; estrogen receptor-positive (ERα^pos^) luminal cells; and estrogen receptor-negative (ERα^neg^) luminal cells. The dotted black line indicates the cross-section of the mammary duct represented in the magnified scheme on the right. This figure was created with Biorender.com (accessed on 12 December 2021).

**Figure 2 biomedicines-10-00003-f002:**
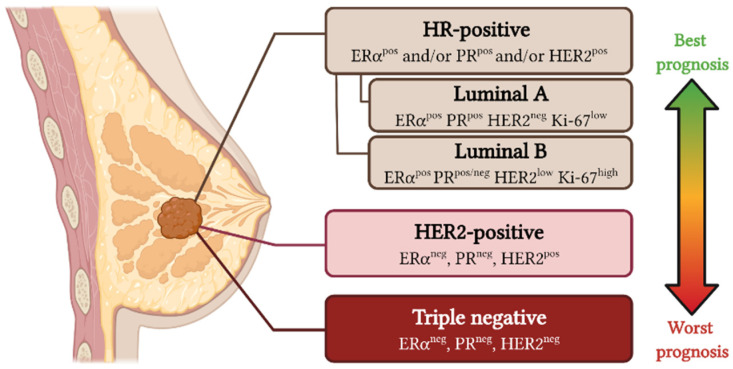
Molecular classification of breast cancer subtypes. Breast cancer can be divided into three main subtypes depending on the histological expression of four markers (ERα, PR, HER2, and Ki-67): hormone receptor (HR)-positive, HER2-positive, and triple-negative. This figure was created with Biorender.com (accessed on 12 December 2021).

**Figure 3 biomedicines-10-00003-f003:**
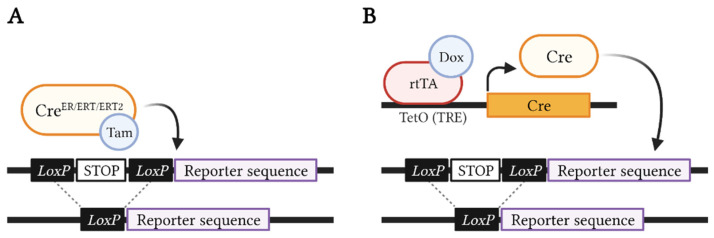
Inducible Cre/Lox systems. (**A**) Tamoxifen-inducible Cre/Lox model, where, upon tamoxifen (Tam) administration, the cells expressing CreER, CreERT, or CreERT2 (determined by a specific promoter) will be labeled with β-galactosidase, fluorescent protein(s), or barcode sequence(s) due to the Cre-mediated recombination of the *loxP* sites, excising the STOP codon in the reporter transgene. (**B**) Tetracycline-inducible Tet/Lox system, where, upon doxycycline (Dox) administration, the cells expressing rtTA will be labeled by the chosen reporter due to the binding of the rtTA transactivator to TRE sequences (such as transcriptional activation elements (TetO)), resulting in Cre expression and consequent excision of the *loxP*-flanked STOP codon upstream to the reporter transgene. This figure was created with Biorender.com (accessed on 12 December 2021).

**Table 1 biomedicines-10-00003-t001:** Murine models that recapitulate different human breast cancer subtypes. This table summarizes the transcriptomic analysis performed in tumors derived from different murine in vivo models that are able to resemble specific human breast cancer subtypes.

Mouse Model	Oncogenic Expression	Human Breast Cancer Subtype	Ref.
*MMTV*-Wnt1	*Wnt1* overexpression	Basal-like	[75]
*WAP*-Tag	SV40 large T antigen	Basal-Like	[74]
*WAP*-Int3	Notch 4 overexpression	Luminal-Like	[78]
*MMTV*-Hras	*Hras* overexpression	Luminal A	[72]
*MMTV*-PyMT	Activation of Src, PI3K, and Shc	Luminal B	[77]
*MMTV*-Neu	Inactivated rat *ErbB2* overexpression in MECs	HER2-enriched	[76]
*WAP*-Cre/Etv6	Etv6-Ntrk3 fusion gene overexpression	HER2-enriched	[82]
Brg1^+/−^	*Brg1* heterozygous	HER2-enriched	[79]
p18^−/−^	*Cdkn2c* homozygous null	HER2-enriched	[80]
Rb^−/−^	*Rb* homozygous null	HER2-enriched	[80]
*MMTV*-Fgf3	*Fgf3* overexpression	HER2-enriched	[83]
*MMTV*-Lpa	*Lpa1*, *Lpa2*, or *Lpa3* overexpression	Mesenchymal-like	[69]
p53^+/−^ irradiated	*Trp53* heterozygous, irradiated	Mesenchymal-like	[71]
DMBA-induced	Random DMBA induction	Basal-like Mesenchymal-like	[68]
C3(1)-Tag	pRb, p107, p130, p53, p300 inactivation and others in MECs	Basal-like Mesenchymal-like	[44]
*WAP*-Myc	*Myc* overexpression in LCs	Basal-like Luminal B	[73]
*MMTV*-Myc	*Myc* overexpression	Basal-like Luminal B	[72]
*MMTV*-Cre/Brca1^Co/Co^/p53^+/−^	*Brca1* truncation in MECs, *Tp53* heterozygous null	Basal-like Mesenchymal-like	[67]
*MMTV*-Aib1	*Aib1* overexpression	Basal-Like Luminal B HER2-enriched	[69]
*WAP*-T_121_	pRb, p107 and p130 inactivation in LCs	HER2-enriched Mesenchymal-like	[70]

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
