# Peer review of "How Lineage Tracing Studies Can Unveil Tumor Heterogeneity in Breast Cancer"

_biomedicines, 2021, doi:10.3390/biomedicines10010003_

Round 1
Reviewer 1 Report
The review by Vinuesa-Pitarch et al discusses how lineage tracking models have been used to identify and track cell of origin for breast cancer. Identifying the cell of origin and the cellular heterogenicity of breast cancers is a highly important topic. This review summarises the studies that have utilised lineage tracing models to study for breast cancer research. It is a succinct review and does cover majority of the current lineage tracing studies, although there have been a few recent articles publish which should be incorporated [Rädler PD, Wagner KU. Sci Rep. 2021; Lee HJ, Jang B. Sci Rep. 2021; Ginzel JD, Snyder JC. Mol Cancer Res. 2021], as well as some these articles [Sikandar SS, Clarke MF. Nat Commun. 2017; Lüönd F, Christofori G. J Mammary Gland Biol Neoplasia. 2019]. The authors discuss the different lineages, mouse models and the type of tumours they form, and whilst the text is succinct, it might be nice to display this graphically for readers. The authors in the introduction use a restricted series of manuscripts to set the tone of the review, namely only referring to the historic ER/PR/Her2/Ki67 classification for breast cancers. Studies using large breast cancer datasets including the study by Dawson/Caldas et al have reported 10 integrative clusters of breast cancer subtypes and these latest classifications have not been mentioned in this review. As diagnosis is moving towards using more -omic technologies then this review should reflect the progression and how lineage tracing is supporting these further in-depth classifications. Whilst the authors do mention the incorporation of scRNA and barcoding technologies in the future, they have limited themselves to only using mouse models. Work carried out by Nguyen and Eaves. Nature. 2015, have barcoded and tracked human breast cells and the types of tumours that arise from these cells. There have been other studies that have used human primary breast cells to label and form tumours and whilst it is a different type of lineage tracing, it is lineage tracing nevertheless and it should be incorporated. They have discussed advantages of using lineage tracing studies to challenge assumed cell line concepts such as EMT which is welcomed and the coverage of cellular plasticity looking at the few different models out there. However, they don’t necessarily delve deeper into how the role of plasticity in breast cancer, i.e. whether the cell of origin or the genetic mutation in the is playing a bigger role development of cancer. Overall, this review is a good summary of the current literature and is useful to the breast cancer field.
Minor concerns
- The authors talk about cancer stem cells, but do not make any mention or refer to any references of CD44. Why not?
- Include that gene expression profiles of normal human breast cells have similarities to different cancer subtypes. Lim et al is not the only study that has carried this out. Others such as Shehata/Stingl 2012, Nguyen/Kessenbrock 2018 etc.
- In section 2. The authors switch between mammary gland references and non-mammary gland references which is a bit confusing. If they are wishing to reference the technology, they should reference the original studies that use a fluorescent protein for cre etc, or alternative only reference studies that have been carried out in the mammary gland. It is confusing to the reader.
- I think reference [29] should be the study by Perou/Botstein 2000, as this was the first study to document the different molecular subtypes of breast cancers.
Reviewer 2 Report
This was a well-organized, easy to read review of a complex field.
Reviewer 3 Report
In this review article, Vinuesa-Pitarch, Ortega-Alvarez and Rodilla discuss the use of experimental lineage tracing studies to investigate not only intra-, but also inter-tumor heterogeneity in mammary tumors (thus, I suggest an appropriate change to the title of the paper). The authors generally do a good job summarizing the pertinent literature.
The major problem is that it has been known for a long time that mouse models are poor representations of human breast cancer. Among other issues, there are marked differences in pathologic features, receptor profile, and pattern of metastasis. The authors acknowledge these limitations in several areas (lines 143-148, 251-253, 316-322). However, they constantly switch back and forth from human breast cancer to experimental models, which makes for confusing reading.
Importantly, what is lacking is a critical evaluation of the applicability of the experimental lineage tracing studies to human breast cancer biology. This deserves at least a couple of paragraphs, and the authors may wish to include references that are particular to studying heterogeneity in human breast cancers, such as single cell gene expression profiling and geospatial profiling (e.g., Nanostring and 10X technologies). It is a big leap of faith to propose that lineage tracing studies in mouse models can lead to changes in therapy for human breast cancers.
The authors also should address the following issues related to their figures and table:
- Figure 1 suggests that there are similar numbers of ER+ and ER- luminal cells. In fact, the great majority of human luminal epithelial cells are ER-. Also, the authors should discuss the relationship/distinction between basal and myoepithelial cells.
- Figure 2 is a bit inaccurate in that human breast cancers that are positive for both hormone receptors and HER2 typically are categorized as Luminal B tumors. The figure also does not include the claudin-low or normal-like subtypes that are mentioned in the text.
- Regarding Table 1, did the authors receive permission to adapt it from a previously published article (Ref #33)?
Round 2
Reviewer 3 Report
Please delete the original version of Figure 2.